# Optimum Water and Nitrogen Management Increases Grain Yield and Resource Use Efficiency by Optimizing Canopy Structure in Wheat

**Yang Liu [1], Mao Yang [1], Chunsheng Yao [1], Xiaonan Zhou [1], Wei Li [1], Zhen Zhang [1], Yanmei Gao [1], Zhencai Sun [1,2], Zhimin Wang [1,2] and Yinghua Zhang [1,2,*]**

[1] College of Agronomy and Biotechnology, China Agricultural University, Beijing 100193, China; liuyyang2021@163.com (Y.L.); 18185975906@163.com (M.Y.); yaoyiyao225@163.com (C.Y.); cau20163010008@gmail.com (X.Z.); wellion@cau.edu.cn (W.L.); zhangsirzz@163.com (Z.Z.); 18235409700@163.com (Y.G.); sunzhencai00@163.com (Z.S.); Zhimin206@263.net (Z.W.)

[2] Scientific Observation and Experiment Station of Wuqiao for Crops with High Water Use Efficiency, Ministry of Agriculture, Cangzhou 061807, China

\* Correspondence: zhangyh1216@126.com

**Abstract:** Excessive nitrogen (N) application rates and serious over-exploitation of groundwater under farmer practice threatens the sustainable use of resources in the North China Plain (NCP). Crop canopy structure affects light distribution between leaves, which is important to determine crop growth. A field experiment conducted from October 2016 to June 2019 in the NCP was designed to examine whether optimum water and nitrogen management could optimize canopy characteristics to improve yield and resource use efficiency. Field treatments included: (1) an example of local farming practices, which include the addition of 330 kg N ha$^{-1}$ and irrigated twice (FP), (2) a reduced N rate of 270 kg N ha$^{-1}$ and irrigated twice (T1), (3) a reduced rate of N rate of 210 kg N ha$^{-1}$ and irrigated once (T2), and (4) no N applied (0 kg N ha$^{-1}$) and irrigated once (T3). Results showed that the highest yield was in T1 treatment during 3 years' winter wheat growing seasons. Water use efficiency (WUE), N use efficiency (NUtE), and N partial factor productivity (PFP$_N$) were significantly higher in T2 treatment than in FP, and the three-year average yield was 9.4% higher than that at FP. Optimum crop management practice (T1 and T2) improved canopy structure characteristics, with less relative photosynthetically active photon flux density (PPFD) penetrated of canopy and higher population uniformity as well as leaf area index, to coordinate the distribution of light within the canopy and maximize canopy light interception, resulting in higher yield and resource use efficiency. Leaf area index (LAI) and specific leaf area (SLA) were closely correlated with grain yield and WUE, while PPFD penetrated of canopy was negatively and significantly correlated with grain yield and WUE. The results indicate that canopy structure characteristics, especially PPFD and population uniformity are good indicators of yield and resource use efficiency.

**Keywords:** wheat yield; canopy structure; nitrogen use efficiency; water use efficiency

## 1. Introduction

The North China Plain (NCP) is a major winter wheat production area in China, producing approximately 60% of the total national wheat production with less than 8% of the total water resource in China [1]. Smallholder farming dominates the agricultural landscape in the NCP. In order to increase the wheat yield, excessive fertilizer application and over-exploitation of groundwater irrigation occurred in the last two decades [2–5]. However, Ray et al. [6] found that about 56% of wheat planting areas in China, including NCP, in which wheat yields were stagnating. At the same time, excessive nitrogen (N) application rates and the overuse of aquifers under farmers' practice reduced the groundwater table, polluted the environment, and threatened sustainable agriculture in the NCP [7,8].

Thus, the major challenge of winter wheat farming in the NCP is to overcome the overuse and low use efficiency of nitrogen (N) fertilizer and water by farmers [9,10].

Agronomists tested many crop management techniques to improve nitrogen use efficiency [11,12]. Many studies have shown that the optimal N fertilizer input (around 185 kg N ha$^{-1}$) could improve NUE without yield losses [8,13,14]. In addition, topdressing N fertilizer and a combination of organic fertilizers were effective in increasing nitrogen use efficiency of wheat [15–18].

In addition to N management, water management is also very important to improve crop yield, as well as water use efficiency (WUE) and nitrogen use efficiency (NUE). Many studies have found that water deficits after anthesis result in early senescence and more remobilization of pre-anthesis stored assimilates to grains in cereals: thus, yield was prone to reduced [19–21]. Although Xu et al. [5] reported that optimal limited irrigation practice could ensure grain yield and substantially increase WUE in the NCP, farmers still are prone to overusing water and N fertilizer to increase yield. However, it is widespread that crop yield and resource use efficiency fail to be further improved under more water and N fertilizer supply. Previous studies had identified the factors that contributed to the farmers' low yield, including management deficiencies, low plant density, unsuitable sowing time, and backward irrigation infrastructure [22,23]. However, the reason of farmers' low yield was not analyzed from the perspective of crop canopy structure.

Canopy structure is potentially an important determinant of the observed functional response. Crop management practices are those improved canopy light and nitrogen distributions to maximize canopy photosynthesis, often resulted in higher yield and NUE [24]. Hence, a better understanding of canopy structure is necessary to accurately quantify the distribution of light and its relationship with grain yield and resource use efficiency.

Plant architecture including the leaf area index (LAI) and plant height indicate light intercepted by plant canopy [25]. Niinemets [26] reported light interception in plant stands from leaf to canopy. At the leaf level, the alterations in leaf chlorophyll content and leaf dry mass per unit area (MA) affects the amount of leaf area and, then, influences the light interception. However, at the canopy level, branching frequency and the allocation of photosynthates to plant leaf determine the rate of light harvesting. N fertilization can also affect light absorption and extinction in the crop plant by affecting crop plant growth [26]. Previous study used the photosynthetic photon flux density (PPFD) attenuation in the canopy to describe the vertical light distribution within a canopy [24]. With the light intensity decreasing from top to bottom in the canopy, the dry mass per unit of leaf area decreased. In other words, the leaf area per unit of leaf weight increased and the leaf becomes thinner in order to receive more light. The tillers rate per stem decreased and the canopy structure became flatter under the less light interception. All these changes affect crop response in low light and, hence, result in a more efficient crop canopy structures to enhance light harvesting [27].

The specific objectives of this study were to (1) compare the yield and resource use efficiency between different nitrogen rates combined with irrigation practices and farmers' practice; (2) investigate the relationships between canopy characteristics and crop yield under different management practices; (3) explore whether the canopy structure can be improved by optimizing cultivation management as a result to improve productivity, WUE and NUE. We believe that the knowledge obtained through this study will gain insight for developing high yield and NUE crop management practice through regulating the canopy structure.

## 2. Materials and Methods

### 2.1. Field Descriptions

A 3 y field experiment was conducted at Wuqiao Experimental Station of China Agricultural University in Hebei Province (37°41′N, 116°36′ E) during the 2016/2017, 2017/2018, and 2018/2019 winter wheat growing seasons. The experimental site is located in the Wuqiao County, as is typical of the North China Plain, which has a warm-temperate

sub-humid continental monsoon climate. Rainfall and daily mean air temperature in the 2016–2017, 2017–2018, and 2018–2019 growing seasons are shown in Figure 1. The soil texture of 100 cm depth of the experimental plot is light loam. The soil bulk density and field capacity in 0–200 cm soil layers are presented in Table 1.

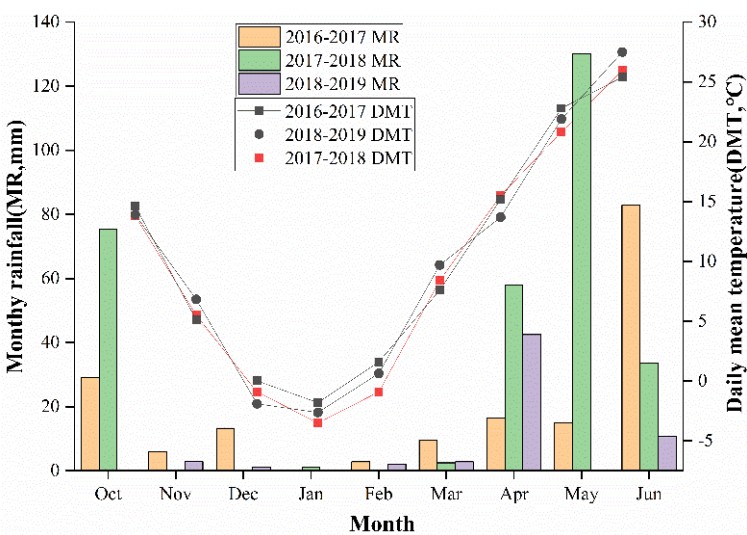

**Figure 1.** Rainfall and daily mean air temperature during the three growing seasons.

**Table 1.** The soil bulk density and field capacity in 0–200 cm soil layers in the experimental plots.

| Soil layer(cm) | 0–20 | 20–40 | 40–60 | 60–80 | 80–100 | 100–120 | 120–140 | 140–160 | 160–180 | 180–200 |
|---|---|---|---|---|---|---|---|---|---|---|
| Bulk density (g cm$^{-3}$) | 1.52 | 1.55 | 1.58 | 1.63 | 1.64 | 1.63 | 1.65 | 1.62 | 1.57 | 1.61 |
| Field capacity (%) | 24.48 | 31.78 | 31.42 | 27.71 | 25.39 | 28.52 | 26.27 | 27.30 | 26.17 | 25.93 |

### 2.2. Experimental Design

In this experiment, farmers' practice (FP) was included among the treatments. In FP, 330 kg N ha$^{-1}$ was applied with ratio of 6:4 at the pre-sowing and jointing stages. One hundred and twenty kilograms of P$_2$O$_5$ and 120 kg K$_2$O ha$^{-1}$ were supplied just before sowing. The plots were irrigated at jointing and anthesis, and each irrigation amount was 75 mm. On the basis of FP, treatment T1 was established, in which N rate was reduced to 270 kg N ha$^{-1}$, and the ratio of N application was adjusted to 5:5 at the pre-sowing and jointing stages. The irrigation mode was similar to FP. Besides, organic manure (15,000 kg ha$^{-1}$) was applied just before sowing to improve the soil quality, 150 kg P$_2$O$_5$ ha$^{-1}$, 150 kg K$_2$O ha$^{-1}$ and micro-nutrients of Zn (30 kg ha$^{-1}$) were also added just before sowing. In treatment T2, N rate was reduced to 210 kg N ha$^{-1}$ with ratio of 6:4 at the pre-sowing and jointing stages, 120 kg P$_2$O$_5$ ha$^{-1}$ and 120 kg K$_2$O ha$^{-1}$ were supplied just before sowing. Only once irrigation 75 mm was applied at jointing to increase resource use efficiency. There was a blank control with no fertilizer application (T3) and once irrigation 75 mm at jointing. There was no P and K fertilizer and other fertilizer application during wheat growing under the four treatments. Compared with FP, T1 treatment was designed to increase yield. T2 treatment with optimized crop management was developed to test the resource use efficiency potential in the North China Plain. Winter wheat were sown in October using a sowing machine set to a row spacing of 15 cm. Four treatments were designed in a block design with four replicates, each replicate was 4 × 10 m. Dates of sowing, jointing stage, anthesis stage, and harvest are listed in detail in Table 2 for the three seasons.

**Table 2.** Dates of sowing, jointing stage, anthesis, and harvest under different treatments in the 2016–2019 growing seasons of winter wheat.

| Year | Treatment | Dates of Sowing | Dates of Jointing Stage | Dates of Anthesis Stage | Dates of Harvest |
|---|---|---|---|---|---|
| | FP | 14 October 2016 | 7 April 2017 | 5 May 2017 | 8 June 2017 |
| 2016–2017 | T1 | 14 October 2016 | 7 April 2017 | 5 May 2017 | 8 June 2017 |
| | T2 | 14 October 2016 | 6 April 2017 | 4 May 2017 | 7 June 2017 |
| | T3 | 14 October 2016 | 6 April 2017 | 4 May 2017 | 7 June 2017 |
| | FP | 22 October 2017 | 11 April 2018 | 5 May 2018 | 8 June 2018 |
| 2017–2018 | T1 | 22 October 2017 | 12 April 2018 | 5 May 2018 | 8 June 2018 |
| | T2 | 22 October 2017 | 10 April 2018 | 4 May 2018 | 7 June 2018 |
| | T3 | 22 October 2017 | 9 April 2018 | 3 May 2018 | 5 June 2018 |
| | FP | 15 October 2018 | 6 April 2019 | 5 May 2019 | 9 June 2019 |
| 2018–2019 | T1 | 15 October 2018 | 6 April 2019 | 5 May 2019 | 9 June 2019 |
| | T2 | 15 October 2018 | 5 April 2019 | 4 May 2019 | 8 June 2019 |
| | T3 | 15 October 2018 | 4 April 2019 | 3 May 2019 | 6 June 2019 |

*2.3. Sampling and Measurements*

2.3.1. Dry Matter, Grain Yield, and Harvest Index

Two 0.5 m inner rows of plants from each plot were cut at ground level at anthesis (Z61) and maturity (Z91) to measure the plant aboveground dry matter. For the samples at maturity, spikes were threshed to determine grain weight, and HI was calculated as the ratio of grain weight to total aboveground dry matter. The grain yield for each plot (with 13% water content) was measured from an area of 4 $m^2$ after maturity. The spike number per $m^2$ was counted in six 2 m inner rows, and the grain number per spike was determined by counting the grains of each spike from 100 randomly selected plants in each plot before harvest. The 1000-grain weight (with 13% water content) was calculated by weighing 1000 seeds from the yield measurement sample with 3 replicates.

Contribution ratio of dry matter after anthesis to grain = (dry matter at anthesis− dry matter at maturity without grain)/grain dry matter.

2.3.2. Water Use Efficiency, N Partial Factor Productivity, and N Use Efficiency (NUtE)

The evapotranspiration (ET) during the growing season of winter wheat was determined using the following soil water balance equation [28]:

$$ET = SI + P + \Delta W \tag{1}$$

where ET (mm) is total evapotranspiration during a growing season; SI (mm) is the irrigation amount; P (mm) is the precipitation; $\Delta W$ (mm) is the soil water storage at sowing minus the soil water storage at maturity at a 2 m soil depth. No runoff and drainage were observed at any of the experimental sites in this study; thus, they were ignored. Capillary rise was also negligible because the groundwater table was lower than 2.5 m below the ground surface [29].

Total nitrogen concentration (Nc) in plants was determined using the Kjeldah method [30]. Water use efficiency (WUE), nitrogen use efficiency (NUtE), and N partial factor productivity (PFPN) were calculated as follows [31,32]:

$$WUE = GY/ET \tag{2}$$

$$PFP_N = GY/N \tag{3}$$

$$\text{Plant-acquired N} = DM \times Nc\% \tag{4}$$

$$NUtE = GY/\text{Plant-acquired N} \tag{5}$$

where GY (kg $ha^{-1}$) is the grain yield, ET (mm) is the total evapotranspiration, N is the nitrogen application rate (kg $ha^{-1}$) applied in the experimental treatments, DM is dry

matter accumulation of plants at maturity; Nc% is the nitrogen concentration in plants at maturity.

2.3.3. Canopy Characteristics (Tillers Rate per Stem, SPAD, PPFD, LAI, and SLA)

In order to clarify the canopy characteristics under different managements, the light distribution in the canopy and photosynthetic traits including chlorophyll content, specific leaf area and leaf area index as well as the tillers rate per stem were measured.

Tillers rate per stem (%) = (spikes number at maturity- emergence number)/emergence number × 100% [33].

Chlorophyll content (SPAD) in the flag, second, and third leaves from the top were measured with a SPAD-502 Minolta chlorophyll meter (Spectrum Technologies, Plainfield, IL, USA) in 10 leaves per plot starting from anthesis until 30 days after anthesis (30DAA).

The relative photosynthetically active photon flux density (PPFD) of the top three leaves and whole canopy intercepted were measured at 0, 3, 9, 15, 21, 24, and 28 days after anthesis for each replication by linear photosynthetic active radiation ceptometer (AccuPAR, Decagon Devices Inc., Washington, USA). All the PPFD measurements were taken when the sky was clear and sunny and restricted to from 11:00 h to 14:00 h (solar time).

Specific leaf area was calculated by dividing the area of a leaf by its dry weight at anthesis; it means leaf area per unit weight.

Leaf area index (LAI) was measured at anthesis. The top three leaves and the remaining green leaves area were measured by a Li-3100 area meter (Li-Cor, Inc., Lincoln, Nebraska, USA), and green leaf area index (LAI) was calculated.

*2.4. Data Analysis*

Figure 1 was created using OriginPro 2016 (OriginLab Corp., Northampton, MA, USA). Other figures were created using software R [34], bars in figures represent the standard errors of mean. Treatment means were compared using the least significant difference test with a significance level of $p < 0.05$. Pearson correlation analysis was conducted by the correlation procedure in the SPSS (IBM SPSS Statistics 24).

**3. Results**

*3.1. Yield*

As shown in Table 3, grain yield was lowest in T3 treatment (0 kg N ha$^{-1}$), and the highest grain yield was achieved in T1 treatment (270 kg N ha$^{-1}$) for 3 winter wheat growing seasons. Grain yield in T1 was significantly higher than FP and T3 treatments, and there was no significant difference in yield between T2 and FP except that T2 was significantly higher than FP in the 2016–2017 growing season. Compared with FP, on average, an 18.9% higher yield was achieved by reducing 18% N rate, changing N rate applied ratio, and increasing organic manure in T1. By optimizing management, grain yield was on average improved by 9.4% in T2 under reducing 36% N rate as compared with FP treatment.

**Table 3.** Grain yield, yield components, and harvest index (HI) under different treatments in the 2016–2019 growing seasons of winter wheat.

| Year | Treatment | Yield (t ha$^{-1}$) | Spike Number (m$^{-2}$) | Grains (spike$^{-1}$) | 1000 Grain Weight (g) | HI |
|---|---|---|---|---|---|---|
| | FP | 8.3 b | 672.2 b | 28 b | 41.2 d | 0.40 b |
| | T1 | 9.5 a | 765.0 a | 33 a | 42.7 c | 0.43 a |
| | T2 | 9.2 a | 701.5 b | 32 a | 44.8 b | 0.44 a |
| | T3 | 7.4 c | 625.2 c | 27 c | 48.1 a | 0.44 a |
| 2016–2017 | FP | 6.4 b | 614.5 b | 27 a | 44.5 b | 0.43 c |
| 2017–2018 | T1 | 7.0 a | 742.7 a | 28 a | 43.5 c | 0.45 bc |
| 2018–2019 | T2 | 6.5 b | 612.6 b | 27 a | 45.6 a | 0.45 b |
| | T3 | 4.4 c | 557.6 c | 21 b | 45.7 a | 0.48 a |
| | FP | 7.4 b | 611.5 c | 27 b | 46.7 a | 0.41 b |
| | T1 | 9.9 a | 813.8 a | 33 a | 42.7 b | 0.43 b |
| | T2 | 8.7 ab | 700.0 b | 34 a | 39.4 c | 0.46 a |
| | T3 | 5.1 c | 495.9 d | 18 c | 46.4 a | 0.43 b |
| Mean | FP | 7.4 b | 632.7 b | 27 b | 44.0 b | 0.42 c |
| | T1 | 8.8 a | 773.9 a | 31 a | 43.0 b | 0.44 b |
| | T2 | 8.1 ab | 671.4 b | 31 a | 43.2 b | 0.45 ab |
| | T3 | 5.6 c | 559.6 c | 22 c | 46.8 a | 0.45 a |

Values within columns followed by the different letters are statistically significant at $p < 0.05$ level among treatments.

Spike number m$^{-2}$ showed the same trend as grain yield. In the highest yield treatment (T1), the spike number was also the highest; however, there was opposite trend between yield and thousand grain weight (TGW). There was no significant difference in grain number per spike between T1 and T2, but they were significantly higher than FP and T3 (Table 3).

### 3.2. Population Uniformity

In Table 4, T1 had the higher plant height (PH) and tillers rate per stem (TRS) than FP; at the same time, it had lower variation coefficients in PH and TRS than FP, indicating that the population in T1 was relatively uniform. There was no significant difference in PH and TRS between T2 and FP; however, T2 had a lower variation coefficient than FP, resulting in a higher population uniformity than FP. There were similar trends in the three growing seasons.

**Table 4.** Plant height and tillers rate per stem (TRS) and respective coefficient of variation (CV) at maturity under different treatments in the 2016–2019 growing seasons of winter wheat.

| Treatment | 2016–2017 | | | | 2017–2018 | | | | 2018–2019 | | | |
|---|---|---|---|---|---|---|---|---|---|---|---|---|
| | Height (cm) | CV1 (%) | TRS (%) | CV2 (%) | Height (cm) | CV1 (%) | TRS (%) | CV2 (%) | Height (cm) | CV1 (%) | TRS (%) | CV2 (%) |
| FP | 74.1 b | 4.8 | 18.3 bc | 19.9 | 69.5 b | 3.4 | 11.3 b | 19.2 | 75.9 b | 3.6 | 18.9 b | 35.7 |
| T1 | 80.0 a | 1.8 | 37.2 a | 14.8 | 74.2 a | 1.3 | 31.1 a | 15.8 | 77.4 a | 1.0 | 38.6 a | 3.0 |
| T2 | 75.0 ab | 4.6 | 22.6 b | 6.6 | 69.0 b | 1.0 | 15.5 b | 12.7 | 73.8 b | 1.4 | 21.3 b | 24.3 |
| T3 | 66.5 c | 2.1 | 11.4 c | 27.5 | 58.0 c | 1.4 | 3.1 c | 26.0 | 56.5 c | 2.1 | 0.9 c | 23.0 |

CV1: the variation coefficient of wheat height; CV2: the variation coefficient of tillers rate per stem. Values within columns followed by the different letters are statistically significant at $p < 0.05$ level among treatments.

### 3.3. LAI, PPFD Penetrated, and SLA

The leaf area index (LAI), specific leaf area (SLA) at anthesis, and relative photosynthetically active photon flux density (PPFD) after anthesis were examined to reflect canopy structure characteristics under different treatments. For 3 years' growing seasons, the LAI of T1 was significantly higher than that of other treatments, and there was no significant difference between T2 and FP (Figure 2).

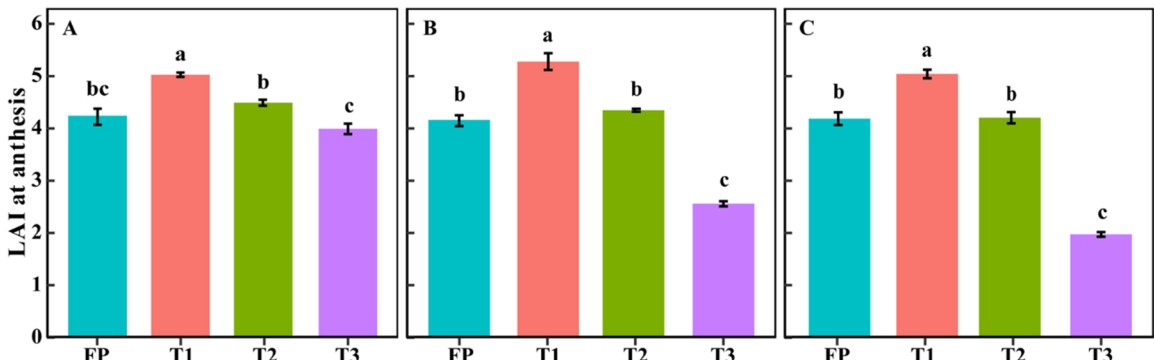

**Figure 2.** Leaf area index (LAI) of total green leaves at anthesis under different treatments in 2016–2017 (**A**), 2017–2018 (**B**), and 2018–2019 (**C**) growing seasons. Different letters in the figure indicate statistical differences among treatments (LSD$_{p < 0.05}$).

The SLA of flag leaf at anthesis under T1 and T2 treatments were lower than that of FP and T3, while the opposite trend was observed in the second leaf and third leaf (Figure 3). It indicated that the flag leaf area of T1 and T2 treatment was thicker and larger than that of other treatments.

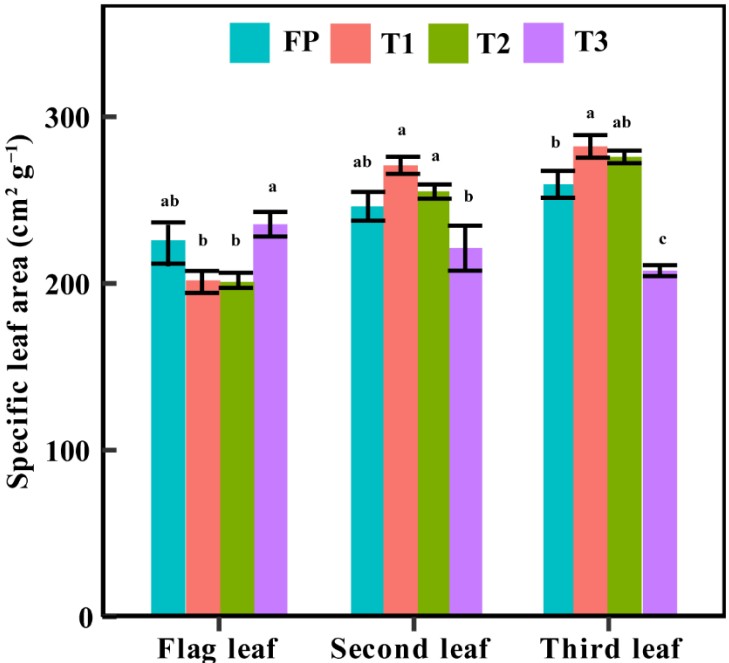

**Figure 3.** Specific leaf area of the top three leaves at anthesis under different treatments in 2017–2018 growing season. Different letters in the figure indicate statistical differences among treatments (LSD$_{p < 0.05}$).

The PPFD can reflect the interception of light radiation. The differences of light radiation interception by top three leaves and whole canopy under different treatments was shown in Figure 4. After flowering, the values of relative PPFD penetrated under the canopy increased during grain filling stage. With the leaves gradually aged, the canopy light interception decreased. The light radiation was intercepted by the top three leaves and whole canopy in the tendency of T1 > T2 > FP > T3.

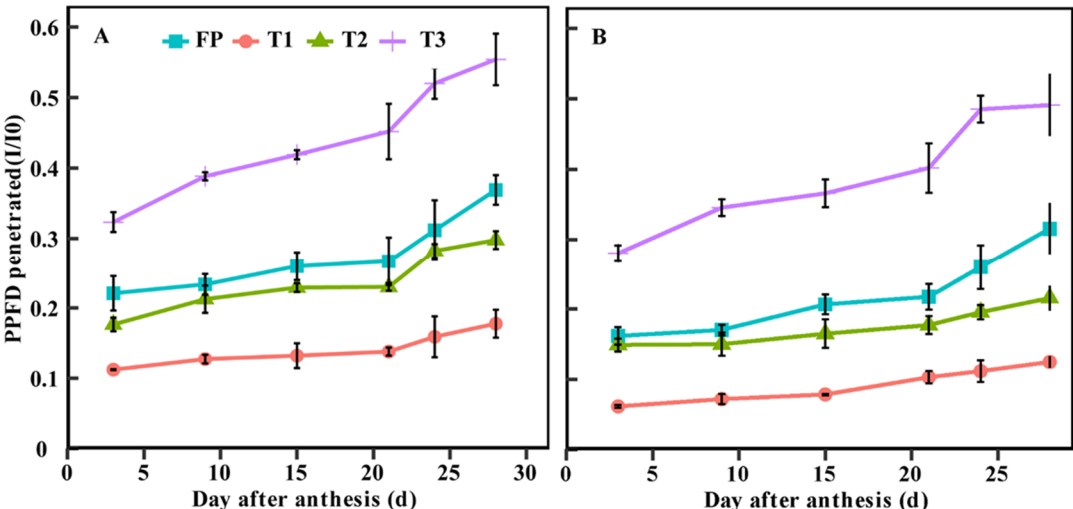

**Figure 4.** Relative photosynthetically active photon flux density (PPFD) penetrated of the top three leaves (**A**) and whole canopy (**B**) at 3, 9, 15, 21, 24, and 28 days after anthesis under different treatments in 2017–2018 growing season.

### 3.4. Chlorophyll Content (SPAD)

In 2016–2017 growing season, the chlorophyll content in the flag and second leaf of T1 was highest after anthesis, and those of FP was similar to T2 treatment; while the chlorophyll content in the third leaf of FP was higher than that of other treatments and it began to decrease until 18 days after anthesis (Figure 5 A–C).

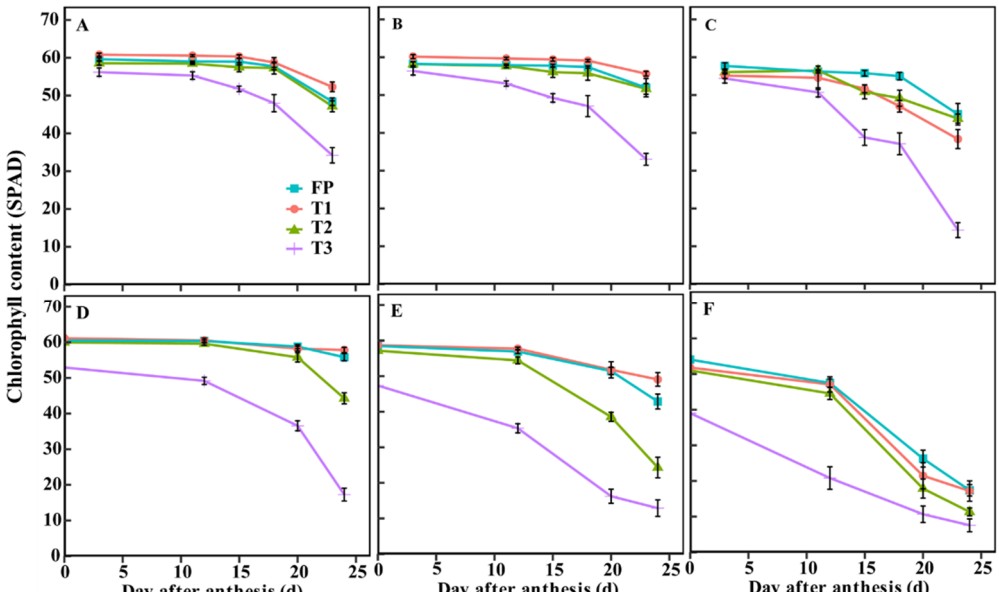

**Figure 5.** Chlorophyll content (SPAD) of the flag leaf (**A** and **D**), the second leaf (**B** and **E**), and the third leaf (**C** and **F**) under different treatments at 3, 11, 15, 18, and 23 days after anthesis in 2016–2017 (**A–C**) and at 0, 12, 20, and 24 days after anthesis in 2017–2018 (**D–F**) growing season.

In the 2017–2018 growing season, the chlorophyll content of the top three leaves in T2 were lower than those in T1 and FP treatments, and in the second leaf of T2, it decreased rapidly after 11 days after anthesis (DAA11). Chlorophyll content in the third leaf of FP was also higher than that of other treatments. There is no nitrogen fertilizer application in T3 and its chlorophyll content in the top three leaves was significantly lower than in other

nitrogen treatments. In a word, increasing the nitrogen rates delayed the leaf senescence, especially for the lower leaf in the canopy.

### 3.5. Dry Matter and HI

As shown in Figure 6, dry matter at anthesis and maturity in T1 treatment was highest in all treatments, and T3 treatment was lowest due to the lack of N fertilizer. There was no significant difference in dry matter at anthesis and maturity between T2 and FP. The harvest index (HI) was lowest in FP treatment (Table 3), and it was always lower than in T2 during the three growing seasons, which may lead to the lower grain yield than T2. The trend of contribution ratio of dry matter after anthesis to grain was T1 > T2 > FP > T3, consistently (Figure 6).

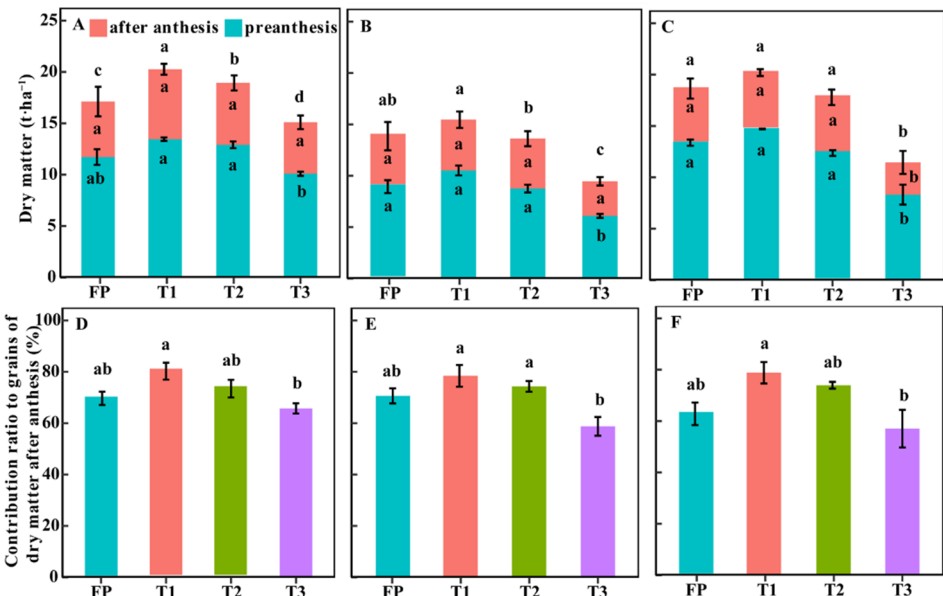

**Figure 6.** Dry matter at anthesis, after anthesis (**A**–**C**) and contribution ratio of dry matter after anthesis to grain (**D**–**F**) under different treatments in 2016–2017 (**A** and **D**), 2017–2018 (**B** and **E**), and 2018–2019 (**C** and **F**) growing seasons. In Figure **A**–**C**, the bottom half of the column is the dry matter pre-anthesis, shown in blue color, while the top half of the column is the dry matter after anthesis, shown in red color. Different letters in the figure indicate statistical differences among treatments ($LSD_{p < 0.05}$).

### 3.6. WUE, NUtE, and $PFP_N$

Across the three winter wheat growing seasons, T1 had the highest ET, with a mean of 480 mm (Table 5). The 3 years' average ET under T2 treatment (401 mm) was significantly lower (by 12.6%) than under FP (459 mm) without yield losses. Meanwhile, T2 saved 75 mm irrigation water every winter wheat growing season. Over the 3-year period, the mean WUE of 1.5 kg m$^{-3}$ in T3 treatment was the lowest among the four treatments (Table 5). The averaged WUE is 1.8 kg m$^{-3}$ in T1 and 2.1 kg m$^{-3}$ in T2, which were 12.5 and 31.3% higher than that of FP (1.6 kg m$^{-3}$), respectively.

**Table 5.** Annual irrigation, precipitation, ΔW, evapotranspiration (ET), and water use efficiency (WUE) under different treatments during 2016–2019 growing seasons of winter wheat.

| Year | Treatment | Irrigation (mm) | Precipitation (mm) | Soil Water Storage at Sowing (mm) | Soil Water Storage at Maturity (mm) | ET (mm) | WUE (kg·m$^{-3}$) |
|---|---|---|---|---|---|---|---|
| 2016–2017 | FP | 150 | 83.5 | 742.2 | 489.7 | 486 b | 1.7 b |
| | T1 | 150 | 83.5 | 775.2 | 492.7 | 516 a | 1.9 a |
| | T2 | 75 | 83.5 | 776.2 | 456.7 | 478 b | 1.9 a |
| | T3 | 75 | 83.5 | 778.5 | 469.0 | 468 b | 1.6 c |
| 2017–2018 | FP | 105 | 187.9 | 797.2 | 659.1 | 431 a | 1.5 b |
| | T1 | 105 | 187.9 | 786.8 | 626.7 | 453 a | 1.5 ab |
| | T2 | 75 | 187.9 | 752.8 | 617.7 | 398 b | 1.6 a |
| | T3 | 75 | 187.9 | 778.9 | 692.8 | 349 c | 1.2 c |
| 2018–2019 | FP | 150 | 62.3 | 770.8 | 522.1 | 461 a | 1.6 c |
| | T1 | 150 | 62.3 | 753.0 | 495.3 | 470 a | 2.1 b |
| | T2 | 75 | 62.3 | 740.5 | 550.8 | 327 b | 2.7 a |
| | T3 | 75 | 62.3 | 706.2 | 517.5 | 326 b | 1.6 c |

ET (mm) is total evapotranspiration during a growing season; soil water storage at sowing and maturity at a 2 m soil depth. Values within columns followed by the different letters are statistically significant at $p < 0.05$ level among treatments.

The detailed information of N uptake, nitrogen use efficiency (NUtE), and N partial factor productivity (PFP$_N$) were listed in Table 6. For N uptake, it was significantly higher in T1 treatment than other treatments. Although the amount of N rates applied in FP was the highest, there was no significant difference in nitrogen uptake between FP and T2. Meanwhile, T2 saved 36% N rates compared with FP. Nitrogen use efficiency (NUtE) was the fraction of plant acquired N to be converted to grain yield (kg kg$^{-1}$). Treatment T3 performed best in NUtE for three growing seasons, because of its low nitrogen uptake. NUtE in T2 was significantly higher than T1 and FP, and NUtE in FP was the lowest. The PFP$_N$ was higher in T2 than in T1 and FP by 18.7 and 73.5% on average, respectively.

**Table 6.** Annual chemical N fertilizer input, total N uptake by the plant, N use efficiency (NUtE), and partial factor productivity from applied N (PFPN) under different treatments during 2016–2019 growing seasons.

| Year | Treatment | N Fertilizer (kg ha$^{-1}$) | N Uptake (kg ha$^{-1}$) | NUtE (kg·kg$^{-1}$) | PFP$_N$ (kg·kg$^{-1}$) |
|---|---|---|---|---|---|
| 2016–2017 | FP | 330 | 256 b | 32.4 c | 25.2 c |
| | T1 | 270 | 270 a | 35.2 bc | 35.2 b |
| | T2 | 210 | 248 b | 37.1 b | 43.8 a |
| | T3 | 0 | 165 c | 44.8 a | / |
| 2017–2018 | FP | 330 | 215 b | 29.8 b | 19.4 c |
| | T1 | 270 | 243 a | 28.8 b | 25.9 b |
| | T2 | 210 | 199 b | 32.7 ab | 30.9 a |
| | T3 | 0 | 118 c | 37.3 a | / |
| 2018–2019 | FP | 330 | 267 b | 27.7 b | 22.4 c |
| | T1 | 270 | 322 a | 30.7 b | 36.7 b |
| | T2 | 210 | 240 b | 36.3 a | 41.4 a |
| | T3 | 0 | 131 c | 38.9 a | / |

Values within columns followed by the different letters are statistically significant at $p < 0.05$ level among treatments.

### 3.7. Pearson Correlation Coefficients among GY, WUE, NUtE, Canopy Characteristic Traits

Grain yield and WUE were significantly correlated with the dry matter at anthesis (DMA), dry matter after anthesis (DMAA) and at maturity (DMM), contribution ratio of dry matter after anthesis to grain (CRG), grain number per m$^2$ (GN), spike number per m$^2$ (SN), total green LAI at anthesis, and flag leaf SPAD at 24 DAA (Table 7). Nitrogen use efficiency (NUtE) was significantly and positively correlated with the 1000-grain weight

and PPFD of whole canopy intercepted at 28 DAA. Grain number per m² and spike number per m² were significantly correlated with total green LAI at anthesis and flag leaf SPAD at 24 DAA.

**Table 7.** Pearson correlation coefficients among GY, WUE, NUtE, canopy characteristics traits agronomical and physiological traits.

|  | DMA | DMM | DMAA | CRG | GN | TGW | SN | LAI | FSPAD | FSLA | PPFD | GY |
|---|---|---|---|---|---|---|---|---|---|---|---|---|
| LAI | 0.80 ** | 0.83 ** | 0.58 ** | 0.74 ** | 0.68 ** | −0.41 * | 0.88 ** | 1 |  |  |  |  |
| FSPAD | 0.59 ** | 0.63 ** | 0.54 ** | 0.71 ** | 0.62 ** | −0.54 ** | 0.67 ** | 0.86 ** | 1 |  |  |  |
| FSLA | −0.54 | −0.57 | −0.39 | −0.52 | −0.56 | 0.36 | −0.60 * | −0.64 * | −0.50 | 1 |  |  |
| PPFD | −0.89 ** | −0.89 ** | −0.52 | −0.71 ** | −0.65 * | 0.57 | −0.78 ** | −0.94 ** | −0.81 ** | 0.61 * | 1 |  |
| GY | 0.96 ** | 0.94 ** | 0.77 ** | 0.53 ** | 0.92 ** | −0.45 * | 0.78 ** | 0.70 ** | 0.53 ** | −0.62 * | −0.90 ** | 1 |
| WUE | 0.62 ** | 0.63 ** | 0.41 * | 0.42 * | 0.70 ** | −0.65 ** | 0.53 ** | 0.67 ** | 0.53 ** | −0.65 * | −0.75 ** | 0.75 ** |
| NUtE | 0.07 | 0.06 | 0.02 | −0.36 | 0.05 | 0.50* | −0.19 | −0.36 | −0.57 ** | −0.38 | 0.74 ** | 0.14 |

DMA, dry matter at anthesis; DMM, dry matter at maturity; DMAA, dry matter after anthesis; CRG, contribution ratio of dry matter after anthesis to grain yield; GN, grain number per spike; TGW, thousand grains weight; SN, spike number per m²; LAI, total green leaf area index at anthesis; FSPAD, flag leaf SPAD at 24 DAA; FSLA, flag leaf specific leaf area at anthesis; PPFD, relative photosynthetically active photon flux density penetrated of whole canopy at 28 DAA; GY, grain yield; WUE, water use efficiency; NUtE, nitrogen utilization efficiency. * and ** mean significant difference at $p < 0.05$ and $p < 0.01$, respectively.

## 4. Discussion

### 4.1. Grain Yield, Dry Matter, and Resource Use Efficiency as Affected by Water and Nitrogen Management

Previous studies have shown that the winter wheat yield and resource use efficiency could be improved by optimizing the management of water and fertilizer on the North China Plain [35,36]. In this study, the grain yield of T1 treatment (8.8 t ha⁻¹) and T2 treatment (8.1 t ha⁻¹) averaged over three years were 18.9 and 9.4% higher than that of FP treatment, respectively (Table 3), and T2 treatment obtained the higher NUtE (35.4 kg kg⁻¹), PFP$_N$ (38.7 kg kg⁻¹), and WUE (2.1 kg·m⁻³) (Tables 5 and 6). This showed that optimizing nitrogen rate and irrigation period can further increase nitrogen use efficiency and WUE in the NCP.

Grain yield in wheat is directly related to the spike number [37]. As Slafer et al. [38] reported that a small variation in yield may be due to either grains per m² or individual grain weight, while grains per m² accounted for large changes in yield. Changes in grains m⁻² are primarily associated with spike number m⁻² [38]. However, the spike number is most vulnerable to environmental conditions and management practices. The water and nitrogen interaction was significant for yield, biomass at maturity, and spike number [39]. In this study, spike number and grains per spike under T1 treatment were significantly higher than FP, whereas there was no significant difference in 1000-grain weight between the two treatments across the three winter wheat growing seasons (Table 3). This showed that optimum water and nitrogen management increased spike number and grain number per spike and, then, increased grain yield. Of course, there was Zn application in T1 treatment, which may also contribute to the increase in grain yield. Yilmaz et al. [40] also tested that biomass production and yield components were increased by soil application of Zn compared with other applications.

Improving biomass production and HI is a highly promising approach to increase grain yield [41,42]. Further, it is believed that, in most situations, increases in dry matter remobilization from vegetative tissues to grains and the harvest index are closely associated with higher grain yields, PFP$_N$ and WUE in cereals [43–45]. In this study, correlation analysis showed that the dry matter at anthesis (DMA), dry matter after anthesis (DMAA) and at maturity (DMM), and contribution ratio of dry matter after anthesis to grain (CRG) were all significantly and positively correlated with grain yield and WUE (Table 7). Although the dry matter at anthesis and maturity of FP was higher than that of T2 during the 2017–2019 two years' growing seasons, the yield was lower than that of T2 because of its lower harvest index (Table 3 and Figure 6). It is probably that water deficiency in T2 promoted pre-anthesis dry matter from vegetative tissues to grains during the filling stage [44] and, thus, increased the harvest index, WUE, and NUtE.

Across the 3 years, compared to FP, the WUE of T1 and T2 increased by 12.5 and 31.3%, respectively (Table 5). The increase in WUE for T1 was due to the higher increase in grain yield (18.9%) than ET (4.6%), while for T2, the increase in WUE was due to the increase in grain yield (9.4%) and the decrease in ET (12.6%). This study also showed that when rainfall was high (2017-2018), the soil water consumption was low; when the rainfall was low (2016–2017 and 2018–2019), the soil water consumption was high. In addition, WUE was significantly and positively correlated with grain yield (Table 7). This result was similar to our previous research [46]. Therefore, grain yield can be used to predict WUE of wheat in the NCP.

*4.2. Canopy Structure as Affected by Water and Nitrogen Management*

Crop management practices can coordinate the distribution of light and N within the canopy by improving canopy eco-physiological characteristics to maximize canopy photosynthesis and result in higher yield and NUE [24]. Research has shown that N fertilization management affects leaf chlorophyll content and, hence, light harvesting by plant [27]. Leaf area index (LAI) affects canopy structure, and it together with chlorophyll contents can diagnose nitrogen [47]. It was verified that plant N uptake was proportional with LAI [48]. In this study, the chlorophyll content (SPAD) and LAI were higher in T1 than in other treatments, and the SPAD of flag leaf was significantly and positively correlated with yield and WUE; however, it was significantly and negatively correlated with NUtE (Table 7). Although the nitrogen content is related to the photosynthetic capacity of leaves, the more N supply, the higher RuBPCase activity and chlorophyll content; however, excessive N fertilization would result in environmental pollution and low N use efficiency [48–50]. The high chlorophyll content in top three leaves especially in the third leaf after anthesis in FP indicated a lot of nitrogen residue in the leaves [51], which decreased the total nitrogen uptake of the plant (Figure 5, Table 6). As a result, there is lower nitrogen use efficiency in FP than other treatments.

Light interception by the canopy is important for crop growth [52]. Compared with FP, T1 and T2 intercepted more radiation in the top three leaves and the whole canopy after anthesis (Figure 4). Additionally, PPFD penetrated was significantly and negatively correlated with LAI (Table 7). These results are consistent with Liu et al. [53]. LAI at anthesis and flag leaf SPAD at 24 DAA (FSPAD) were significantly and positively correlated with grain yield and WUE, while flag leaf SLA at anthesis and PPFD penetrated of the whole canopy intercepted at 28 DAA were significantly and negatively correlated with grain yield and WUE; the opposite results were found in NUtE (Table 7). Overall, improving the light distribution in the canopy can significantly increase canopy photosynthesis, yield potential, and NUE [24].

It is believed that, in most situations, improving the uniformity of the population is beneficial to intercept radiation [54]. Meanwhile, light interception by the canopy is an important factor to determine dry matter production and crop growth [55]. In this study, the coefficient variance of wheat height and tillers rate per stem (TRS) under FP was higher than that of T1 and T2 (Table 4), which indicates that population uniformity of FP was lower than T1 and T2. T1 intercepted more radiation with better uniformity of the population and, hence, improved dry matter production and yield. T2 also had better population uniformity and canopy structure, resulting in higher yield and resource use efficiency.

There are many factors affecting population uniformity and canopy structure in the farmers' practice (FP). Many farmers were using wheat seeds, which were harvested last year and sowing next growing season to save seed cost. In addition, poor quality of sowing results in a shortage of seedlings. Canopy structure significantly affects crop light interception and photosynthesis and further influences productivity [24]. The reason for farmers' low yield was analyzed from the perspective of crop canopy structure. Overall, our research demonstrated that those managements that can increase grain yield and resource use efficiency all improved canopy structure.



## 5. Conclusions

Adoption of integrated management practices, i.e., reducing N fertilizer rates, improving irrigation management, and applying organic fertilizer, could increase both grain yield and resource use efficiency. Compared with FP, the average irrigation and N rate of the T2 treatment were significantly lower, whereas WUE, PFP$_N$, and NUtE in T2 were significantly higher than those of FP. Increases in yield under T1 and T2 treatments were mainly through improving population uniformity and further coordinating distribution of light within the canopy. Canopy structure profile parameters, i.e., the variation coefficient of tillers rate per stem and plant height, relative photosynthetically active photon flux density (PPFD) penetrated of whole canopy, LAI and SLA are responsible for canopy productivity. Besides, T2 treatment with one-time irrigation at jointing increased the contribution ratio of dry matter after anthesis to grain and, thus, improved the harvest index, which contributed to higher grain yield and higher resource use efficiency. The effect of a single irrigation event on grain yield in winter wheat growing seasons under different rainfall conditions needs to be further studied in the future.

**Author Contributions:** Data curation, Y.L.; Formal analysis, Z.S. and Y.Z.; Funding acquisition, Z.W. and Y.Z.; Investigation, Yang Liu, M.Y., C.Y., Z.Z. and Y.G.; Methodology, Y.L., X.Z. and Z.W.; Software, W.L.; Writing—original draft, Y.L., Z.W. and Y.Z.; Writing—review & editing, Z.W. and Y.Z. All authors have read and agreed to the published version of the manuscript.

**Funding:** This research was funded by the [National Key Research and Development Program of China] grant numbers [2016YFD0300105, 2016YFD0300401], the National Natural Science Foundation of China (31871563), and the Earmarked Fund for Modern Agro-Industry Technology Research System (CARS-3).

**Data Availability Statement:** The data presented in this study are available on request from the corresponding author.

**Acknowledgments:** This study was funded by the National Key Research and Development Program of China (grant numbers: 2016YFD0300105, 2016YFD0300401), the National Natural Science Foundation of China (31871563), and the Earmarked Fund for Modern Agro-Industry Technology Research System (CARS-3).

**Conflicts of Interest:** The authors declare no conflict of interest.

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
