# Peer review of "Optimum Water and Nitrogen Management Increases Grain Yield and Resource Use Efficiency by Optimizing Canopy Structure in Wheat"

_agronomy, doi:10.3390/agronomy11030441_

Round 1
Reviewer 1 Report
Dear authors
The manuscript is well written but can further be improved by addressing these concerns attached. Thank you.

Author Response
Thanks for your valuable comments!

Reviewer 2 Report
Excellent paper and good piece of research.
I have a few requests below for clarification.
Firstly the 2 abstracts have some differences. The abstract above has been condensed slightly and states that FP was 300 kg/ha N not 330 kg/ha N as stated in the paper. Which is right and please change the wrong one?
In materials and methods, there is no mention of the soil type that the trials are grown on. Table 1 is the only reference to soil and includes some very detailed depth measurements of %field capacity. The paper is about water and N management. Soil type is critical to this and so too is water movement in the soil. These both need to be shown. Not just a static reference to water capacity in table 1 but the change in soil water from beginning to end of each year of crop growth.
Therefore instead of table 1 a more informative table needs to be made which shows each of the determinants in equation 1 that have been used to calculate the evapotranspiration ET. This has then been used in table 4 to work out the water use efficiency. The measurements have been done so no more work is needed other than to show the values of each of the 4 determinants used for equation 1 during each of the 3 years of crop growth. The only determinant given is irrigation. The others that need to be shown in table form are precipitation, soil water storage at sowing and at harvest for each of the 3 years of the experiment.
On page 3 under experimental design, 15 t/ha OM was applied to T1. The paper needs to say when this was done and whether it was just the once. Also since this is a paper looking at the resource use efficiency of adding both water and N then it would be useful to know how much N is in the manure and what proportion is available to the crop.
Also it just needs mentioning if any other nutrients apart from Zn were applied to any of the treatments. If none then say so.
Under experimental design it needs stating how big the trial plots were and if there were replicates, how many? This information is needed if someone wants to repeat the experiment.
Page 4 under ‘results’, the authors say ‘reducing N rate by 10%’ from FP to T1 suggests that if T1 is 270 kg/ha then FP is 300 kg/ha and not 330. See my comment at the beginning.
Page 10 at the bottom, the authors say that flag leaf SLA at anthesis was significantly and positively correlated with grain yield GY and WUE, while flag leaf SLA at anthesis was significantly and negatively correlated with GY & WUE. Both statements can’t be true. In fact looking at table 6, the second statement is true not the first. This needs correcting.
Page 11 at end of discussion just needs some English improving, ‘Our research results demonstrated that crop management practices that improve canopy structure and population uniformity, which would increase yield and resource use efficiency’ ie its not all crop management practices that improve canopy structure but those that do will increase yield and resource efficiency.
The final sentence of the conclusion backs up what I was saying above in paragraph 3 about showing how ET is calculated. This paper gives 3 years of ‘different rainfall conditions’ yet no data is shown. What I’ve recommended above will aid the ‘further discussion’ that the paper mentions in looking at the effect of a single irrigation event.
So it’s not just a case of ‘ET phone home’, to quote the famous film but ET needs more clarification with how its worked out as stated in paragraph 3 above.
Author Response
Thanks for your valuable comments!

Reviewer 3 Report
This article contains both major scientific errors and minor editing wording issues. These might be missing due to writing style and not the actual experiment, however I am unable to assess the value of the experiment because details are missing in the writing.
Abstract:
Abstract contains many grammatical errors, wording that is confusing, and very long complex sentences. Please rewrite for clarity. Example:
“The field experiments included four treatments: (1) farmers’ practice (FP) in local production were introduced with 330 kg N ha−1 and two times irrigation, (2) treatment T1 was established on the basis of FP with reduced N rate to 270 kg N ha−1, (3) treatment T2 with 210 kg N ha−1 and one irrigation and (4) treatment T3 with 0 kg N ha−1 and one irrigation.”
to
Field treatments included: 1) an example of local farming practices which include the addition of 330 kg Nha-1 and irrigated twice, 2) a reduced N rate of 270 kg N ha-1 and irrigated twice, 3) a reduced rate of N rate of 210 kg N ha-1 and irrigated once, and 4) no N applied (0 kg N ha-1) and irrigated once.
First sentence grammatical correction for “overuse of underground water under farmers’ practice threaten sustainable agriculture”.
Introduction:
A better literature review is required. Please provide a better connection between WUE and NUE with plant canopy cover in small grains.
Methods:
How much estimated rainfall occurred during the experiment?
Experimental design is missing information such as plot size and experimental design (how many reps and how was is laid out).
Treatment T1 describes rate of N of 270 kg, but also states manure was added (which contains N), so what is the total application of the plot?
In T1 Zn is added. Was Zn added to any additional treatments? This addition will make interpretation of data difficult.
What was the timing of the irrigation applications (i.e. plant development stage) and the reasoning for those timings?
Define what is meant by “optimized crop management” either in introduction or within the paragraph.
In the sampling and measurements section write out the abbreviation “HI”
Write dry matter after anthesis to grain ratio as a formula.
How was plant acquired N measured?
Section 2.3.3 should be written in paragraph format where applicable. An explanation sentence should be added to explain why these measurements are important to the study (i.e. The following traits were measured to describe plant canopy structure).
Cite Least significant difference test.
Results:
Table 2 – organize treatments (rows) in logical that matches your description in the methods, same for all the figures.
Figure 5. - unclear how the dry matter is separated. Please clarify in figure legend.
Discussion:
In general, the lack of information in the methods does not help the reader understand how the authors came to conclusions about the results. Example: the conclusions of optimized N rate and irrigation period leading to increased N efficiency and WUE is unsubstantiated or unclear due to the cofactors of manure and Zn additions of T1.
Many of the conclusions lack support from other sources causing the reader to wonder how the authors came to those conclusions. This study may be unique for the geographical area, but it is not unique for the subject matter (i.e. the relationship between NUE, yield and WUE). A more complete search of the literature is required. Example: “The high chlorophyll content in top three leaves after anthesis in FP indicating much nitrogen residue in the leaves”. How is that statement supported by other research?
Author Response
Thanks for your valuable comments!

Reviewer 4 Report
-Initial soil test maybe include if available
-Annual rainfall and temperature data for the experimental season should be added in a table format
Author Response
Thanks for your advice, rainfull and daily mean air temperature of 3 years’ growing seasons are shown in Fig. 1. I have added “The soil texture of 100 cm depth of the experimental plot is light loam. ” in “2.1. Field descriptions”. I am very sorry that I did not test the initial soil properties.
Round 2
Reviewer 1 Report
Dear Authors,
The manuscript has greatly improved and ready for publication after a few grammatical errors are addressed.
For example in the abstract, change ‘overexploitation’ to ‘over exploitation’, the word threatened, to ‘threatens’.
Please add the word ‘are’ between ‘practices’ and ‘those improved’ in this statement found in the introduction.
“Crop management practices those improved canopy light and nitrogen distributions to maximize canopy photosynthesis, oftenand resulted in higher yield and NUE
[24].”
In Figure Figure 3. Specific leaf area of the top three leaves at anthesis under different treatments in 2017-
2018 growing season. Please lable A, or B to indicate the graph for 2017, then for 2018. In addition, check and make the x-axis labels (Flag leaf, Second leaf, Third Leaf) of the top graph visible.
Thank you.
Author Response
Thanks for your advice!
1 We have changed ‘overexploitation’ to ‘over exploitation’, the word threatened, to ‘threatens’(Page1, line1 and line2)
2 In the introduction, I have added the word ‘are’ between ‘practices’ and ‘those improved’ . (Page2, line21)
3 In this sentence “Crop management practices those improved canopy light and nitrogen distributions to maximize canopy photosynthesis, oftenand resulted in higher yield and NUE [24].”, the wrong word “ofenand” was changed to “often”.( Page2, line22)
4 There is only one graph in figure 3. Maybe there was something wrong with the "tracking changes" function of my Microsoft Word, which caused the deleted pictures to be displayed. I have modified it in the article.
Thank you.
Reviewer 3 Report
This version of the paper is greatly improved. I still have some minor questions.
What is the experimental plot size (i.e. whole field or subplots within field). If subplots how are they distributed or selected in the farmer’s field? How many subplots? Are the plots used for yield the subplots for all the traits?
What are the sampling numbers for canopy characteristics?
Clearly tie last paragraph regarding population uniformity to introduction. The comment regarding farmers could be added to the argument of why to study canopy structure.
There are minor grammatical issues throughout the document, such as fragment sentences and incorrect usage of pronouns at the beginning of sentences. These should be fixed before publication.
Author Response
Thanks for your advice! The detailed reply to the review is in the attachment, please check
